# Prevalence and determinants of medicinal plants utilization during labour among women of reproductive age in Butiama, Tanzania: A community-based cross-sectional study

Magnus Michael Sichalwe[1,2]*, Nangi William Nangi[1], Leah Daniel[3], JohnPaul Otuomasiri Egbobe[2], Grace Tavengana[2]

1 Division of Health, Social Welfare and Nutrition Services, Butiama District Council, Mara, Tanzania, 2 School of Public Health, Nantong University, Nantong, Jiangsu, China, 3 Division of Health, Social Welfare and Nutrition Services, Mara Region Council, Mara, Tanzania

* sichalwemagnus@gmail.com, sichalwe.magnus@fwpps.org

## Abstract

### Background

The use of certain medicinal plants during childbirth has been linked to negative outcomes such as uterine rupture and foetal distress, both globally and in Sub-Saharan Africa including Tanzania. Despite this, little is known about the factors influencing women's use of medicinal plants during labour or delivery in Tanzania. This study sought to assess the prevalence and determinants of medicinal plants use during labour and/or delivery in the Butiama district.

### Methodology

This community-based quantitative study used a cross-sectional design with 398 participants, selected through multistage sampling. Data were collected via a structured questionnaire in Swahili language using the Kobo Toolbox from June to July 2024. Analysis was performed with SPSS version 27.0, including checks for completeness before data entry. Descriptive statistics were computed for univariate analysis, while bivariate analysis, conducted through cross-tabulation, determined relationships between variables. Multivariate logistic regression identified significant predictors at $p < 0.05$.

### Results

In a study of 398 participants, 233(58.5%) reported using medicinal plants during labour and/or delivery. Peasants and homemakers had 2.6 times higher odds of using medicinal plants than those in formal employment (AOR = 2.584, 95% CI: 1.249–5.349, p = 0.011). Women with one child were 1.8 times more likely to use medicinal plants than those with two or more children (AOR = 1.823, 95% CI: 1.136–2.926,

**Data availability statement:** All relevant data are within the paper and its Supporting information files.

**Funding:** The author(s) received no specific funding for this work.

**Competing interests:** The authors have declared that no competing interests exist.

p = 0.013). Women within five kilometres of a health facility had 47.7% lower odds of using medicinal plants compared to their counterparts (AOR = 0.523, 95% CI: 0.334–0.819, p = 0.005). Married and cohabiting women were 42.1% less likely to use medicinal plants than divorced/separated/widowed women (AOR = 0.579, 95% CI: 0.338–0.990, p = 0.046). Women with fewer than four antenatal visits were 55.6% more likely to use medicinal plants compared to those with four or more visits. (AOR = 0.556, 95% CI: 0.365–0.848, p = 0.006).

## Conclusion

Over fifty percent of participants reported using medicinal plants during childbirth, with socio-economic status and healthcare access factors suggesting that targeted education and interventions around medicinal plant use would be beneficial.

## Introduction

The World Health Organization (WHO) and various studies report that nearly 50% of people in industrialized countries use both conventional and alternative therapies, with rates reaching up to 80% in many developing countries [1,2]. Recent global statistics reveal that each year, about 280,000 women lose their lives due to complications related to pregnancy and childbirth, with the majority of these tragic cases occurring in sub-Saharan countries [3]. Medicinal plants use has significantly increased over the past 20 years [4,5]. Studies have recorded 75 plants used to induce labour, including oxytocic herbs like the African spider plant *(Cleome gynandra),* Ginger *(Zingiber officinale), Cissampelos mucronata,* Mwanamphepo (Cissus species), Ampelocissus, and Cyphostemma [6–9]. These herbs during labour can induce intense, prolonged contractions misaligned with cervical dilation, leading to complications like uterine rupture, postpartum haemorrhage, fetal distress, and an increased risk of cesarean section [10,11]. These issues contribute to higher maternal and neonatal mortality rates [12,13].

Medicinal plants used during early labour can expose both mother and newborn to harmful substances, posing risks [14]. Despite this, many believe it can speed up labour, ease pain, aid in placenta removal, and strengthen uterine muscles after delivery [15]. Factors such as social status, ethnicity, cultural practices, and regional differences influence its use during pregnancy and childbirth [16]. Medicinal plants are typically consumed orally or applied rectally, vaginally, or topically [17].

Debates on the efficacy of medicinal plants during pregnancy and childbirth are often subjective and lack solid evidence [6]. In Sub-Saharan Africa, many medicinal plants remain botanically unidentified and under-researched, raising concerns about potential harm to the fetus from natural toxins crossing the placental barrier [18]. Rigorous studies are needed to evaluate the safety and efficacy of medicinal plants, providing evidence-based guidance for expectant mothers and healthcare providers.

Tanzania faces high perinatal mortality, with 39 deaths per 1000 pregnancies, a neonatal mortality rate of 25 per 1000 live births, and a maternal mortality ratio of

556 per 100,000 live births [19]. Many of these deaths could be prevented through better clinical care during pregnancy, childbirth, and postpartum [20].

Documented medicinal plants used to induce labour in Tanzania, Ethiopia, and the Middle East include ginger (*Zingiber officinale*), onions (*Allium cepa*), and Neem (*Azadirachta indica*) [8,11,14]. In Tanzania, the Reproductive Health Survey in Kigoma revealed that 10.9% of pregnant women used local herbs during pregnancy or labour [21], while studies in Mwanza and Tabora found that 23% and 60% of pregnant women, respectively, used medicinal plants during pregnancy and/or labour [14,22].

In Tanzania, discussions on medicinal plants use are seldom part of routine antenatal care (ANC), often left to healthcare providers' discretion. This community study assessed the prevalence of medicinal plants use during labour and/or delivery among women of reproductive age in the Butiama district, identifying the socioeconomic factors that influence this practice.

## Materials and methods

### Study design

The study employed a community-based analytical cross-sectional design, which was chosen for its practicality, cost-effectiveness, and time efficiency in data collection. This design also facilitated the simultaneous comparison of multiple variables.

### Study settings

The Butiama District Council is one of nine local government authorities in the Mara region of Tanzania, established by a government gazette on May 8, 2013. It is bordered by Tarime district to the north, Serengeti to the east, Bunda to the south, and Lake Victoria to the west. According to the 2022 Tanzania National Census, the population of the Butiama District was 314,880, including 65,899 infants and 130,389 women of childbearing age [23]. The council operated 51 health facilities, including one district hospital, four health centers, and 46 dispensaries. These comprised 39 government-owned, four faith-based, three parastatal-owned and four privately owned facilities. Of these, 43 provided Reproductive and Child Health (RCH) services, 41 offered delivery and family planning services, and two specialized in Comprehensive Emergency Obstetric and Newborn Care (CEmONC).

### Study population

The study included women of reproductive age (15–49) living in the Butiama district who had given birth at least once between 2023 and 2024.

### Sample size determination

Fischer's equation, mentioned in reference [24], was used to calculate the sample size based on a study in the Tabora municipality, Tanzania, which found that 60% of pregnant women used medicinal plants during pregnancy or labour [22]. Fischer's formula $N = \frac{Z^2 \times PQ}{D^2}$.

Where; Z=Alpha risk expressed in Z score

  d=Absolute precision
  P=Prevalence from the previous study
  q=1-P
  P=60%=0.6
  Q=1-0.6 = 0.4
  D=0.05
  Z=1.96

$$N = \frac{1.96^2 \times 0.6 * 0.4}{0.05^2} = 369$$

The initial sample size for the study was 369. To account for a 10% non-response rate, the sample size needed to be adjusted. After factoring in the non-response rate, the new adjusted sample size became 406.

## Sampling technique

Butiama district was conveniently selected to assess the determinants of medicinal plants utilization during labour among women of reproductive age. A multistage sampling technique was employed to select the respondents, involving three steps:

**First stage:** Systematic random sampling was applied to select five out of the 18 administrative wards in Butiama Council, namely Butiama, Butuguri, Sirorisimba, Bisumwa, and Buhemba.

**Second stage:** Within each of the selected wards, simple random sampling was used to choose three villages, leading to a total of 15 villages included in the study.

**Third stage:** In each identified village, about 27 households were systematically sampled. If a household lacked women of reproductive age, sampling continued until a suitable household was found. In households with multiple eligible women, participants were selected through simple random selection. The target sample size was 406, but 398 individuals ultimately participated due to non-responses.

## Inclusion criteria

The study involved women aged 15–49 who had given birth at least once between January 2023 and May 2024 in selected villages and who were willing to consent to participate.

## Exclusion criteria

In this study, we set clear exclusion criteria to ensure safety and ethical conduct. Women who were very ill or medically unstable, regardless of the cause, were excluded to prioritize their well-being. Additionally, women with severe mental illnesses that impaired their understanding of the study or ability to give informed consent were also excluded to ensure that all participants could fully comprehend and voluntarily decide to participate.

## Main outcome variable

The primary outcome of interest was the use of medicinal plants during labour and/or delivery, indicating whether or not a participant reported using such remedies. During data collection, participants were asked about their use of medicinal plants during labour and/or delivery, defined as any plant-based substances consumed in raw or minimally processed form without a prescription or recommendation from a healthcare provider. Common examples included pumpkin roots *(Cucurbita pepo),* African spider plant *(Cleome gynandra),* ginger *(Zingiber officinale Roscoe),* and concentrated green or black tea *(Camellia sinensis).*

For this study, labour and/or delivery were defined as the period from the onset of regular, painful uterine contractions indicating active labour until the birth of the baby and delivery of the placenta. This definition was communicated to participants to clearly distinguish it from earlier stages of pregnancy and ensure accurate reporting of medicinal plant use during childbirth.

## Explanatory (contextual) variables

A range of participant characteristics was determined to describe potential factors associated with medicinal plants. These included age, education level, socioeconomic status, parity, distance to health facilities, access to healthcare services,

previous birth experiences, and the presence of a support system during pregnancy and/or childbirth. These variables were assessed to provide context to the patterns of medicinal plants use during labour and/or delivery.

## Study tool

The questionnaire comprised three sections: (I) sociodemographic characteristics; (II) factors related to medicinal plant use during labour; and (III) perceptions and beliefs regarding these plants. Initially developed in English, the tool was translated into Swahili for clarity and cultural relevance. Validity was assessed through a two-step process involving five experts in public health, maternal and child health, and linguistics. Each item was rated on a 4-point scale (1 = not relevant to 4 = highly relevant). The Item-level Content Validity Index (I-CVI) all items scored an I-CVI of 0.90. The Scale-level CVI (S-CVI), the average of I-CVI scores was 0.93. This slight increase above 0.90 reflects rounding and perfect agreement (I-CVI = 1.00) on several items.

Reliability testing yielded a Cronbach's alpha of 0.84, confirming good internal consistency. Based on these results, participants' perceptions (Section III) were categorized using a five-point Likert scale (like, strongly disagree to strongly agree) to facilitate interpretation, particularly in relation to Table 4. Data collection was conducted between June and July 2024.

## Training of the research assistants

Three Swahili-fluent research assistants with prior data collection experience underwent a two-day training. Day one covered the study's objectives, ethical considerations, and participants' rights. Day two focused on data collection techniques, confidentiality, and rapport building to ensure accurate and respectful interactions.

## Pre-testing of data collection instruments

The interviewer-administered questionnaire was pretested with 10 participants from non-selected villages in Butiama District to identify issues with clarity, sequencing, or cultural relevance. Based on the feedback, the tool was revised to enhance reliability and validity. Pretest responses were excluded from the final analysis. Given the diverse educational backgrounds and literacy levels among participants, a standardized data collection approach was adopted to ensure inclusivity and consistency. All participants, regardless of literacy level, were interviewed face-to-face using the same structured questionnaire. Trained research assistants, fluent in both Kiswahili and the local dialect, administered the questionnaire orally. They were also trained to explain terms such as "medicinal plants" using locally relevant examples to enhance comprehension and minimize misinterpretation. This method ensured that participants who could not read or write were not disadvantaged and that responses were as accurate and comparable as possible across the sample.

Reasons for using medicinal plants during labour were defined as follows and explained to participants during data collection to ensure consistency:

1. Labour induction: Refers to the use of medicinal plants with the intent to initiate uterine contractions before spontaneous labour begins, typically to trigger childbirth.

2. Labour acceleration: Refers to the use of medicinal plants to speed up the progress of labour once it has already started, aiming to shorten the duration of delivery.

3. To relieve pain: Refers to the use of medicinal plants to reduce or manage the intensity of labour pain, serving as a traditional form of analgesia.

4. To keep the fetus healthy: Refers to the belief that medicinal plants help protect or enhance fetal well-being during labour, possibly by preventing perceived complications.

5. Other (like to prevent constipation and vomiting): Includes additional uses such as managing gastrointestinal discomforts during labour, like preventing nausea, vomiting, or constipation.

## Statistical analysis

Before analysis, the data were entered, cleaned, and checked for missing information using Excel, then analyzed using IBM's Statistical Package for the Social Sciences (SPSS) version 27.0. Socio-demographic characteristics were analyzed using descriptive statistics, which indicated frequencies and percentages. Descriptive statistics were employed to assess perceptions toward using local medicinal plants among women of reproductive age. Cross-tabulations determined associations between socio-economic variables using a liberal p-value threshold of <0.25 as a screening criterion [25]. Though not for final inference, this approach, supported by statistical literature and expert forums like Stack Exchange, helps retain variables with potential predictive or confounding value for robust multivariate analysis [26,27]. These variables were analyzed using multivariate logistic regression to identify predictors of medicinal plant use, with statistical significance set at $p \leq 0.05$. For clarity in interpreting adjusted odds ratios (AORs), an AOR statistically represents a positive association when it exceeds 1, its 95% confidence interval (CI) lies entirely above 1, and $p \leq 0.05$; a negative association when it is below 1, its 95% CI lies entirely below 1, and $p \leq 0.05$.

## Ethical consideration

Ethical approval for this study was sought from the Institutional Review Board (IRB) of Muhimbili University of Health and Allied Sciences (reference number MUHAS-REC-06-2024-635). Additionally, an approval letter was requested from the District Medical Officer to conduct the study in their respective administrative territories. Before the study commenced in each village, a courtesy call was made to the concerned Village Executive Officer. Written informed consent was obtained from all participants. For those under 18, assent was obtained and consent was provided by a parent, guardian, or spouse, per local ethical guidelines. All participants under 18 had prior childbirth experience and were married or in informal unions.

## Inclusivity in global research

Additional details on the ethical, cultural, and scientific considerations related to inclusivity in global research are provided in the Supporting Information (S1 Checklist).

## Results

### Demographic characteristics and bivariate analysis of medicinal plants use

A total of 406 participants were initially approached for the study, with 398 providing informed consent, yielding a response rate of 98.03%. The average age of participants was 25.67 years (SD±7.21 years), with 56.3% of participants aged between 15–24 years, and 72.1% being married or cohabiting. Regarding education, 68.3% had completed primary school, while 71.4% had a household income below 100,000 Tzs (36.81 USD). Farming or homemaking was the main economic activity for 65.3% of participants. A total of 67.8% lived over 5 km from a health facility, 61.1% had fewer than four ANC visits, 78.9% delivered in health institutions, 10.1% experienced under-five child loss, and 61.8% were in the lowest wealth index category (Table 1).

Bivariate analysis showed medicinal plants use during labour was higher among women with household monthly income below 100,000 Tzs (69.5%, p=0.019), those living over 5 km from a health facility (62.2%, p=0.004), and those attending fewer than four ANC visits (66.1%, p=0.014). It was also more common among women with lower education levels (65.2%, p=0.094). Variables with p<0.25 were selected for multinomial analysis to identify predictors of medicinal plants use during labour and/or delivery (Table 1).

### Prevalence, types, and reasons for the use of medicinal plants during childbirth

Table 2 shows that out of 398 participants, 233 (58.5%) used medicinal plants during childbirth, with African spider plants (67.0%) being the most commonly used. The main reasons for use were to hasten labour (59.2%). However, 31.3% of users reported suspected adverse effects associated with medicinal plants, including postpartum haemorrhage (PPH),

**Table 1. Demographic characteristics and bivariate analysis of medicinal plants use (n = 398).**

| Variables | Frequency (N) | Percent (%) | Medicinal plants utilization | | P-value |
|---|---|---|---|---|---|
| | | | Yes n (%) | No n (%) | |
| **Age group (years)** | | | | | |
| 15-24 | 224 | 56.3 | 130(55.8) | 94(57.0) | 0.305 |
| 25-34 | 116 | 29.1 | 64(27.5) | 52(31.5) | |
| 35-45 | 58 | 14.6 | 39(16.7) | 29(14.6) | |
| **Marital status** | | | | | |
| Single | 72 | 18.1 | 36(15.5) | 36(21.8) | 0.230 |
| Married/Cohabiting | 287 | 72.1 | 175(75.1) | 112(67.9) | |
| Divorced/Separated/widow | 39 | 9.8 | 22(9.4) | 17(10.3) | |
| **Education of participant** | | | | | |
| None/primary incomplete | 73 | 18.3 | 51(21.9) | 22(13.3) | 0.094 |
| Primary education | 272 | 68.3 | 152(65.2) | 120(72.7) | |
| Secondary school and above | 53 | 13.3 | 30(12.9) | 23(13.9) | |
| **Household monthly income in TZs.** | | | | | |
| <100,000 | 284 | 71.4 | 162(69.5) | 122(73.9) | 0.019 |
| 100,000-500,000 | 85 | 21.4 | 59(25.3) | 26(15.8) | |
| >500,000 | 29 | 7.3 | 12(5.2) | 17(10.3) | |
| **Participant's main economic activity** | | | | | |
| Peasant/homemaker | 260 | 65.3 | 146(62.7) | 114(69.1) | 0.016 |
| Self-employed/entrepreneur | 51 | 12.8 | 62(26.6) | 25(15.2) | |
| Employed in the formal sector | 87 | 21.9 | 25(10.7) | 26(15.8) | |
| **Education of the partner** | | | | | |
| None/primary incomplete | 70 | 17.6 | 35(15.0) | 35(21.2) | 0.157 |
| Primary education | 262 | 65.8 | 162(65.8) | 100(60.6) | |
| Secondary school and above | 66 | 16.6 | 36(15.5) | 30(18.2) | |
| **Distance to the nearest health facility** | | | | | |
| ≤5 | 128 | 32.2 | 88(37.8) | 40(24.2) | 0.004 |
| >5 | 270 | 67.8 | 145(62.2) | 125(75.8) | |
| **Number of ANC visits** | | | | | |
| <4 | 243 | 61.1 | 154(66.1) | 89(53.9) | 0.014 |
| ≥4 | 155 | 38.9 | 79(33.9) | 76(46.1) | |
| **Number of children born to the participant** | | | | | |
| One | 131 | 32.9 | 69(29.6) | 62(37.6) | 0.053 |
| Two to five | 178 | 44.7 | 116(49.8) | 62(37.6) | |
| >5 | 89 | 22.4 | 48(20.6) | 41(24.8) | |
| **Loss of a child below five years old** | | | | | |
| Yes | 40 | 10.1 | 27(11.6) | 13(7.9) | 0.225 |
| No | 358 | 89.9 | 206(88.4) | 152(92.1) | |
| **Place of delivery for recent pregnancy** | | | | | |
| Institution delivery | 314 | 78.9 | 177(76.0) | 137(83.0) | 0.089 |
| Home delivery | 84 | 21.1 | 56(24.0) | 28(17.0) | |
| **Wealth Index** | | | | | |
| Lowest | 246 | 61.8 | 149(63.9) | 97(58.8) | 0.093 |
| Second | 63 | 15.8 | 29(12.4) | 34(20.6) | |
| Middle | 61 | 15.3 | 35(15.0) | 26(15.8) | |
| Highest | 28 | 7.1 | 20(8.6) | 8(4.8) | |

**Table 2.** Utilization and sources of medicinal plants during labour and/or delivery (n = 398).

| Variables | Frequency (N) | Percent (%) |
|---|---|---|
| **Utilization of medicinal plants during childbirth** | | |
| Yes | 233 | 58.5 |
| No | 165 | 41.5 |
| **Type of medicinal plant used during childbirth** | | |
| Pumpkin roots (*Cucurbita pepo*) | 55 | 23.6 |
| African spider plant (*Cleome gynandra*) | 156 | 67.0 |
| Ginger (*Zingiber officinale Roscoe*) | 6 | 2.6 |
| Concentrated green or black tea (*Camellia sinensis*) | 16 | 6.8 |
| **Reasons for using medicinal plants during labour** | | |
| To hasten labour | 138 | 59.2 |
| To keep the foetus healthy | 66 | 28.3 |
| To relieve pain | 21 | 9.1 |
| Other (prevent constipation & vomiting) | 8 | 3.4 |
| **Suspected adverse effects or complications from using medicinal plants during labour and/or delivery** | | |
| Yes | 73 | 31.3 |
| No | 160 | 68.7 |
| **Source of information on using medicinal plants during labour** | | |
| Family members/relatives or friends | 128 | 54.9 |
| Traditional birth attendants | 61 | 26.1 |
| Community health workers | 35 | 15.0 |
| Other | 9 | 3.9 |

birth asphyxia, caesarean section, premature rupture of membranes, and foetal distress. The primary sources of information were family members/friends (54.9%).

## Predictors of medicinal plants utilization among study participants

Table 3 shows married/cohabiting women were 42.1% less likely to use medicinal plants than divorced/separated/widowed women (AOR = 0.579, 95% CI: 0.338–0.990, p = 0.046). Peasants/homemakers had 2.6 times higher odds (AOR = 2.584, 95% CI: 1.249–5.349, p = 0.011) and self-employed/entrepreneurs had 1.9 times higher odds (AOR = 1.862, 95% CI: 1.070–3.240, p = 0.028) of using medicinal plants compared to those employed in the formal sector. Women living within five kilometres of a health facility had 47.7% lower odds of using medicinal plants than those residing farther away (AOR = 0.523, 95% CI: 0.334–0.819, p = 0.005). Additionally, women who attended fewer than four ANC visits had 55.6% more likely to using medicinal plants compared to those with four or more visits (AOR = 0.556, 95% CI: 0.365–0.848, p = 0.006). Women with only one child were 1.8 times more likely to use medicinal plants than those with two or more children (AOR = 1.823, 95% CI: 1.136–2.926, p = 0.013).

## Perceptions and beliefs regarding medicinal plants use during childbirth

Table 4 shows that 48.7% of participants strongly agreed or agreed that medicinal plants are readily accessible in their locality, while 44.5% strongly disagreed or disagreed that they are safer than modern treatments. Similarly, 44.5% strongly agreed or agreed that medicinal plants effectively manage labour pain, whereas 55.5% strongly agreed or agreed that they trust modern medicine for safer childbirth. Notably, 64.8% strongly agreed or agreed that they felt uninformed about the risks of using medicinal plants during labour and/or delivery.

**Table 3. Multivariable logistic regression of predictors of medicinal plants use during childbirth in Butiama, Tanzania (N = 398).**

| Variables | Medicinal plants use | COR 95% C.I. | | P-value | AOR 95% C.I. | | P-value |
|---|---|---|---|---|---|---|---|
| | | Lower | Upper | | Lower | Upper | |
| **Marital status** | | | | | | | |
| Single | 36 (15.5) | 1.294 [0.591–2.833] | | 0.519 | 0.664 [0.292–1.509] | | 0.328 |
| Married/Cohabiting | 175 (75.1) | 1.562 [0.930–2.626] | | 0.092 | 0.579 [0.338–0.990] | | **0.046*** |
| Divorced/Separated/Widow | 22 (9.4) | Reference | | | Reference | | |
| **Education of the participant** | | | | | | | |
| None/primary incomplete | 51 (21.9) | 1.777 [0.849–3.719] | | 0.127 | 0.591 [0.265–1.318] | | 0.199 |
| Primary education | 152 (65.2) | 0.971 [0.516–1.758] | | 0.923 | 1.096 [0.561–2.140] | | 0.788 |
| Secondary school and above | 30 (12.9) | Reference | | | Reference | | |
| **Household monthly income** | | | | | | | |
| < 100,000 | 162 (69.5) | 0.585 [0.349–0.982] | | 0.043 | 1.191 [0.727–1.953] | | 0.488 |
| ≥ 100000 | 71 (30.5) | Reference | | | Reference | | |
| **Participant's main economic activity** | | | | | | | |
| Peasant/homemaker | 146 (62.7) | 0.388 [0.189–0.796] | | 0.010 | 2.584 [1.249–5.349] | | **0.011*** |
| Self-employed/entrepreneur | 62 (26.6) | 0.516 [0.305–0.873] | | 0.014 | 1.862 [1.070–3.240] | | **0.028*** |
| Employed in the formal sector | 25 (10.7) | Reference | | | Reference | | |
| **Education of the partner** | | | | | | | |
| None/primary incomplete | 35 (15.0) | 0.833 [0.425–1.635] | | 0.596 | 1.197 [0.603–2.378] | | 0.607 |
| Primary education | 162 (65.8) | 1.350 [0.783–2.328] | | 0.280 | 0.675 [0.386–1.178] | | 0.167 |
| Secondary school and above | 36 (15.5) | Reference | | | Reference | | |
| **Distance to the nearest health facility** | | | | | | | |
| Below or equal to five kilometres | 88 (37.8) | 1.897 [1.217–2.956] | | 0.005 | 0.523 [0.334–0.819] | | **0.005*** |
| Above five kilometres | 145 (62.2) | Reference | | | Reference | | |
| **Number of ANC visits** | | | | | | | |
| < 4 | 154 (66.1) | 1.665 [1.106–2.506] | | 0.015 | 0.556 [0.365–0.848] | | **0.006*** |
| ≥ 4 | 79 (33.9) | Reference | | | Reference | | |
| **Number of children born** | | | | | | | |
| One | 69 (29.6) | 0.626 [0.373–1.051] | | 0.076 | 1.823 [1.136–2.926] | | **0.013*** |
| Two to five | 116 (49.8) | 0.595 [0.375–0.943] | | 0.027 | 1.524 [0.9–2.582] | | 0.117 |
| > Five | 48 (20.6) | Reference | | | Reference | | |
| **Loss of a child below five years old** | | | | | | | |
| Yes | 27 (11.6) | 0.653 [0.326–1.306] | | 0.228 | 1.499 [0.741–3.034] | | 0.260 |
| No | 206 (88.4) | Reference | | | Reference | | |
| **Delivery place for last pregnancy** | | | | | | | |
| Institution delivery | 177 (76.0) | 1.548 [0.934–2.566] | | 0.090 | 0.598 [0.356–1.006] | | 0.053 |
| Home delivery | 56 (24.0) | Reference | | | Reference | | |
| **Wealth Index** | | | | | | | |
| Lowest | 149 (63.9) | 2.931 [1.125–7.639] | | 0.028 | 0.326 [0.124–0.858] | | 0.023 |
| Second | 29 (12.4) | 1.801 [1.031–3.145] | | 0.039 | 0.520 [0.295–0.915] | | 0.023 |
| Middle | 35 (15.0) | 1.578 [0.777–3.208] | | 0.207 | 0.653 [0.318–1.344] | | 0.247 |
| Highest | 20 (8.6) | Reference | | | Reference | | |

COR - Crude Odd Ratio, AOR - Adjusted Odd Ratio Asterisk (*) indicates statistical significance at p < 0.05.

**Table 4. Perceptions and beliefs of participants regarding medicinal plants use during childbirth (n = 398).**

| Item | Strong Agree | Agree | Neutral | Disagree | Strong disagree |
|---|---|---|---|---|---|
| If medicinal plants are readily accessible in the locality | **110 (27.6)** | **84 (21.1)** | 83 (20.9) | 51 (12.8) | 70 (17.6) |
| A belief that medicinal plants are safer than modern treatments during childbirth. | 44 (11.1) | 101 (25.4) | 76 (19.1) | **115 (28.9)** | **62 (15.6)** |
| Medicinal plants are more effective than conventional medicine in managing labour pain. | **64 (16.1)** | **113 (28.4)** | 80 (20.1) | 102 (25.6) | 39 (9.8) |
| Trust the safety of modern medical interventions during labour. | 50 (12.6) | 97 (24.4) | 76 (19.1) | 118 (29.6) | 57 (14.3) |
| A belief that modern medical treatments ensure the safest childbirth. | **96 (24.1)** | **125 (31.4)** | 59 (14.8) | 87 (21.9) | 31 (7.8) |
| Advising another woman to use medicinal plants for pain relief or faster childbirth. | 50 (12.6) | 107 (26.9) | 76 (19.1) | 104 (26.1) | 57 (14.3) |
| Cultural beliefs shape your view of medicinal plants' effectiveness in childbirth. | 45 (11.3) | 131 (32.9) | 79 (19.8) | 93 (23.4) | 50 (12.6) |
| Healthcare providers are knowledgeable about medicinal plants safety during childbirth. | 64 (16.1) | 116 (29.1) | 80 (20.1) | 99 (24.9) | 39 (9.8) |
| Having sufficient information about the risks of using medicinal plants during childbirth. | 19 (4.8) | 82 (20.6) | 39 (9.8) | **142 (35.7)** | **116 (29.1)** |

Bolded values indicate the most frequent responses, reflecting participants' dominant views.

## Discussion

This community-based, cross-sectional study investigated the prevalence and determinants of medicinal plants utilization during labour and/or delivery among women of reproductive age in Butiama District, Tanzania. The key predictors identified include marital status, occupation, proximity to healthcare facilities, number of ANC visits, and number of children.

This study found that 58.5% of women used medicinal plants during labor and/or delivery, a figure similar to the 60% reported in postnatal clinics in Tabora, Tanzania [22]. However, it is notably higher than the 10.9% in Kigoma, 23% in Mwanza, Tanzania, and 12% in Kenya [14,21,28], but lower than the 70% utilization rate reported in Bangladesh [29]. These differences may stem from sociocultural and ethnic factors, as well as variations in access, costs, and regulations related to herbal and modern medicine [18,30]. Furthermore, participants reported using medicinal plants to hasten labour, a practice likely rooted in cultural beliefs regarding their effectiveness in accelerating childbirth [31]. Similar studies in Sub-Saharan Africa also report that women use herbs for labour assistance, relying on traditional knowledge [6,8,11]. Community education on herbal remedy risks during labour, collaboration with traditional birth attendants, and improved healthcare access via mobile clinics can promote safer childbirth and timely interventions for complications.

This study found women who were married or living with a partner were less inclined to use medicinal plants during pregnancy than those who were separated, divorced, or widowed. Similarly with an Ethiopian study [31]. Married or cohabiting women may have better access to formal healthcare through partner support, reducing reliance on medicinal plants [32]. Conversely, in Zambia, single women were more likely to seek modern healthcare due to greater autonomy and access [33]. In contrast, separated, divorced, or widowed women may experience stress, financial challenges, or social isolation, prompting them to seek alternative remedies [31]. Societal or family expectations may also encourage married women to use modern healthcare, while single women might turn more to traditional treatments [34]. However, a study in Uganda found that women in marital unions were more likely to use medicinal plants than their single counterparts [35]. Tailored educational initiatives should aim to educate women outside of marital unions about the risks of medicinal plants during childbirth. Additionally, involving community leaders, women's partners, elder family members, and others in discussions about medicinal plants can strengthen support systems for women during pregnancy and/or childbirth, promoting a more comprehensive approach to maternal care.

Women engaged in peasant farming or homemaking were more inclined to use medicinal plants during pregnancy, likely influenced by limited healthcare access, financial constraints, lower health awareness, and strong cultural traditions [18]. Their preference for traditional remedies is often driven by affordability and familiarity. Similar studies in Sub-Saharan Africa associate medicinal plants use with financial limitations, accessibility, and cultural traditions [6,18,30]. Some studies

show no significant link between occupation and medicinal plants use, suggesting that personal preferences, dissatisfaction with formal healthcare, or the integration of herbal remedies may be more influential [31]. Enhancing community education on the safe use of medicinal plants, expanding access to affordable healthcare, and incorporating traditional healers into primary healthcare services could promote safer and more informed choices.

Distance to health facilities and ANC attendance are likely interrelated, as women living farther away may miss ANC visits due to access barriers, which could increase reliance on medicinal plants. Although analyzed separately, the overlap between these factors should be acknowledged when interpreting the findings.

Proximity to health facilities appears to promote positive health-seeking behaviours, likely driven by easier access to professional care and reliable health information [18,36]. In this study, women residing within five kilometres of a health facility were less likely to use medicinal plants during pregnancy. These findings are in line with similar research conducted in Tanzania and Ethiopia [22,37]. Long distances to facilities increase travel costs, time, and effort, discouraging women from seeking formal care during pregnancy and/or childbirth, while transportation barriers and low health literacy further limit access in rural areas, as seen in studies from Bangladesh and Zambia [33,38]. Conversely, a study in Ethiopia found that pregnant women living less than 5 km from a health facility were twice as likely to use herbal products compared to those living farther away [39]. In Ethiopia, the widespread acceptance of traditional medicine, rooted in diverse cultures, may lead women near healthcare facilities to use herbal remedies as complementary treatments, influenced by cultural beliefs and healthcare accessibility [31,40], highlighting varying factors that shape medicinal plants use in different regions [18,36]. Health officers should educate communities on the risks of medicinal plants during childbirth and equip healthcare workers to address cultural beliefs while promoting safe practices.

Lower engagement with antenatal care, reflected by fewer than four visits, was associated with increased reliance on medicinal plants among pregnant women. An Ethiopian study found similar results [37]. Women with fewer or no ANC visits may have limited information, increasing their reliance on herbal remedies. Studies indicate that regular visits improve access to information on safe pregnancy practices [41]. Having ANC visits is linked with women's knowledge about the effects of medicinal plants utilization [42]. To improve ANC attendance, the Department of Health can train community and healthcare workers on medicinal plants risks during labour, provide reliable information at ANC clinics, and enhance access through mobile clinics, community visits, radio campaigns, and transport support.

Women with only one child were more likely to use medicinal plants than those with several children, possibly due to limited experience, anxiety, financial challenges, and greater reliance on traditional advice [43]. A study in the Central Appalachian Region found that women who actively seek information are more likely to use herbal remedies [44] Cultural norms in rural areas often support medicinal plants use for pregnancy-related issues, particularly where access to modern healthcare is limited [45]. Conversely, a study in Madagascar found that larger families rely more on traditional medicines [46]. The variations in studies reflect a common trend where women in rural areas, influenced by cultural beliefs, experiences, and economic limitations, rely more on traditional remedies due to limited access to modern healthcare [31]. Local health authorities should educate first-time mothers about medicinal plants risks, enhance healthcare access, and improve awareness through mobile clinics and community campaigns.

Our study found that nearly half of respondents view medicinal plants as easily accessible, reflecting rural preferences linked to limited access to modern healthcare, consistent with previous studies [18,35]. Geographical barriers shape health-seeking behaviours, leading women to choose easily accessible remedies. However, even in urban areas, cultural beliefs are associated with medicinal plants use despite modern treatment availability, highlighting the influence of both location and tradition [47]. Maternal health interventions should address geographic accessibility and cultural beliefs, ensuring modern healthcare access while integrating respectful approaches to traditional practices.

This study showed many respondents were skeptical about the safety of medicinal plants and trusted modern childbirth treatments more. This suggests growing awareness of herbal risks and increased confidence in formal healthcare.

Conversely, a 2018 Bangladesh study reported that 68.6% preferred medicinal plants due to concerns about hospital births [29]. This divergence suggests that safety perceptions are heavily influenced by personal experiences and societal narratives rather than empirical data [48]. The concerns about safety in our study, despite a 58.6% usage rate, may stem from personal experiences or community anecdotes. Cultural beliefs and societal narratives influence perceptions, as women may value medicinal plants for its cultural significance but remain cautious about its safety [49]. Additionally, regional differences in healthcare experiences affect how women view the risks of both medicinal plants and modern treatments. This highlights the need for targeted education to address misconceptions and empower women to make informed health decisions.

This study revealed that nearly half of the participants perceived medicinal plants as more effective for managing labour pain, a view similarly reported in a study from Northwest Ethiopia [37]. However, a meta-analysis indicated that while some alternative methods may provide relief, they often lack robust clinical evidence compared to modern interventions [50]. The perception of effectiveness highlights the necessity for further study into the efficacy, specifically, and safety of medicinal plants used during childbirth. Establishing a body of evidence supporting or refuting these beliefs can guide healthcare practices and patient education.

Most of women in our study lacked awareness of the risks associated with medicinal plant use during childbirth, a pattern also observed in studies from Malawi, Tanzania, and a recent meta-analysis. [6,22,36]. One study reported that women who received structured education about the risks and benefits of various childbirth methods felt more empowered and made more informed choices [37]. The information gap underscores the need for comprehensive health education programs tailored to women's specific needs regarding medicinal plants and modern medical options. Empowering women with knowledge can enhance their decision-making capabilities and potentially improve maternal health outcomes.

## Study limitations

While this study provides valuable insights through its rural, community-based design, several limitations should be considered. The main limitations include the cross-sectional design, which limits causal inference and may be affected by residual confounding, and the potential for recall and social desirability bias despite mitigation efforts through structured and confidential interviews. Additionally, participants may have interpreted the concept of "medicinal plants" differently based on personal or cultural beliefs, which could affect response consistency. Finally, although the sample size was sufficient for primary analyses, generalizability to broader populations may be limited.

## Conclusion

This community-based study reveals that 58.5% of participants in Butiama District, Tanzania, use medicinal plants during labour and/or delivery, indicating a significant prevalence of this practice. Key factors influencing this practice include marital status, education level, distance to health facilities, and number of children. Recognizing medicinal plants in maternal healthcare is crucial, and integrating education on traditional remedies into formal healthcare could improve maternal health outcomes. Addressing gaps in knowledge about the risks and benefits of both herbal and modern treatments is essential for empowering women to make informed health choices. This study brought timely attention to the importance of speaking about the use of medicinal plants as we provide maternal health services.

## Recommendations

To enhance maternal health outcomes in Butiama District, we recommend the following actions:

1. Health facility authorities should establish guidelines to incorporate discussions on medicinal plant use into ANC services.

2. Targeted health education programs should raise awareness among women of reproductive age about the benefits and risks of both herbal and modern medicine, addressing existing misconceptions.

3. Given widespread mobile phone access, the Ministry of Health and partners should develop culturally appropriate educational content delivered through mobile platforms.

4. To improve healthcare access in remote areas, mobile clinics should be deployed, and community health workers trained to provide essential care and health education.

5. Engaging community leaders and influencers is vital to promote positive health-seeking behaviours during pregnancy and childbirth.

## Supporting information

**S1 Dataset. Dataset of medicinal plants utilization among women of reproductive age.**
(XLSX)

**S1 File. Data Collection Tool: Questionnaire on the utilization of medicinal plants among women of reproductive age.**
(DOCX)

**S1 Checklist. Summarizes actions taken to ensure diversity, equity, and representation in the study's design and implementation.**
(DOCX)

## Acknowledgments

We extend our gratitude to the Butiama District Executive Director and Medical Officer for granting permission to conduct this study in their district. We also thank the women who volunteered to participate, as well as the Ward and Village Executive Officers for their support in facilitating the research within their areas.

## Author contributions

**Conceptualization:** Magnus Michael Sichalwe.

**Data curation:** Magnus Michael Sichalwe.

**Formal analysis:** Magnus Michael Sichalwe, Nangi William Nangi, Grace Tavengana.

**Funding acquisition:** Magnus Michael Sichalwe.

**Investigation:** Magnus Michael Sichalwe.

**Methodology:** Magnus Michael Sichalwe, Nangi William Nangi, JohnPaul Otuomasiri Egbobe, Grace Tavengana.

**Project administration:** Magnus Michael Sichalwe, Nangi William Nangi, Leah Daniel.

**Resources:** Magnus Michael Sichalwe.

**Software:** Magnus Michael Sichalwe.

**Supervision:** Magnus Michael Sichalwe, Nangi William Nangi, Leah Daniel.

**Validation:** Magnus Michael Sichalwe.

**Visualization:** Magnus Michael Sichalwe.

**Writing – original draft:** Magnus Michael Sichalwe, Nangi William Nangi, Leah Daniel, JohnPaul Otuomasiri Egbobe, Grace Tavengana.

**Writing – review & editing:** Magnus Michael Sichalwe, Nangi William Nangi, Leah Daniel, JohnPaul Otuomasiri Egbobe, Grace Tavengana.

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
