## [Decision Letter · Decision Letter 0]

24 Jan 2025

PONE-D-24-53690
Factors influencing herbal medicine utilization during labour and delivery among women of reproductive age in Butiama, Tanzania: A community-based cross-sectional study
PLOS ONE

Dear Dr. SICHALWE,

Thank you for submitting your manuscript to PLOS ONE. After careful consideration, we feel that it has merit but does not fully meet PLOS ONE’s publication criteria as it currently stands. Therefore, we invite you to submit a revised version of the manuscript that addresses the points raised during the review process.

The manuscript has been assessed by two reviewers, and their comments are appended below.

We look forward to receiving your revised manuscript.

Kind regards,

Jianhong Zhou

Staff Editor

PLOS ONE

Journal Requirements:

2. We note you have included a table to which you do not refer in the text of your manuscript. Please ensure that you refer to Table 4 in your text; if accepted, production will need this reference to link the reader to the Table.

3. Please remove all personal information, ensure that the data shared are in accordance with participant consent, and re-upload a fully anonymized data set. 

Reviewers' comments:

Reviewer's Responses to Questions

**Comments to the Author**

1. Is the manuscript technically sound, and do the data support the conclusions?

Reviewer #1: Yes

Reviewer #2: Partly

2. Has the statistical analysis been performed appropriately and rigorously? 

Reviewer #1: Yes

Reviewer #2: No

3. Have the authors made all data underlying the findings in their manuscript fully available?

Reviewer #1: Yes

Reviewer #2: Yes

4. Is the manuscript presented in an intelligible fashion and written in standard English?

Reviewer #1: Yes

Reviewer #2: No

5. Review Comments to the Author

Reviewer #1: This is an interesting paper about use of herbal medicine in a Butiama, Tanzania during labour and delivery among women of reproductive age. The study is well done, I underline some brief points:

1. The title does not reflect the entire content of the article; it needs to be reviewed.

2. Table 1 should be eliminated because the majority of the information has been repeated in Table 2. The latter is largely sufficient to present the results.

3. Please correct in the Page 10, table 1 the value, ‘’never used HM = 165’’ et Utilization of HM during childbirth, No = 164) ?

4. Table 1 : this question is not correct ‘’ Type of HM used during childbirth’’, because the proposed answers are varied.

5. In Table 2, the sum of the values for all the variables studied does not give 233, which is the total of participants who confirmed the use of plants. All the values in this table must be corrected.

Reviewer #2: Dear Authors,

Thank you for the possibility to read your interesting paper.

Assessing the statistics is challenging. I downloaded the Excel-file and imported it to SPSS to check some data and calculations.

First of all, I found that there seems to be several errors in Table 2. For “Age group” the percentages are mixed up (% for “all participants” come under “No”, % for “Yes” belong to “No” etc). This made me do the analysis in SPSS and I get different values for both counts and p-values for most of the factors in the table. I struggle to see why you have chosen a significance level of 0.25. that is very high. And talking about significance, it is difficult to see what is significantly higher or lower than what.

For Table 3 many confidence intervals are overlapping – does that mean anything for the results?

Table 4 has some numbers in bold and marked with stars – what do those markings refer to? They normally refer to significant values, but if that is the case here, what method is used for the analysis? Remember that a table should always be understandable in itself so this one needs some sub-text with explanations.

Other comments:

A definition of “herbal medicine” would be useful. How did you explain to the participants what you were asking about? And is it necessary to make the abbreviation “HM”?

Where does reference 1 say that “conventional and alternative therapies contribute to 80% of global healthcare”?

You say that “Studies show that herbal medicine during labour induces intense, prolonged contractions….etc…” but which herbal medicines? It can not be a general statement about herbal medicines. Reference 8 mentions one herb.

What does the number of modes of administration have to do with the cultural significance of herbal medicine?

Your section about training research assistants is rather detailed.

Ethics: I guess you mean “Muhimbili University of Health and Allied Sciences”.

Results:

Response rate: This is not about how many you aimed to include but about how many you asked to participate versus how many who were willing.

Please reconsider how you write the sentences with numbers and percentages – it does not flow as it is now.

I suppose you KNOW how many participants used herbal medicines during childbirth – not “around 233” (which is an approximate number) but “233”. Same goes for “About 272” participants with primary education – it is 272 – not “about”.

Why do you mention “roots” and “leaves” but not which roots or leaves? Does this matter? Is it interesting? Why?

Discussion:

The statement “Women’s desire to maintain mariages,….. reliance on HM” requires a reference.

Significance of educational level: there is a significant difference between those with none (reference) and those with primary education, but what can you say about those with primary vs. those with secondary education? Overlapping confidence intervals – does that matter? Confidence intervals including “1” – does that matter?

Please also revise abstract accordingly.

6. PLOS authors have the option to publish the peer review history of their article (what does this mean?). If published, this will include your full peer review and any attached files.

Reviewer #1: No

Reviewer #2: No

---

## [Author Response · Author response to Decision Letter 1]

14 Feb 2025

Dear Reviewers,

We would like to express our sincere gratitude for your valuable feedback and thoughtful comments on our manuscript. Your constructive suggestions have significantly contributed to enhancing the quality and clarity of our study. We truly appreciate the time and effort you dedicated to reviewing our work. Each of your points has been carefully addressed in detail, and we have uploaded a full set of responses in the rebuttal letter for your review.

We believe that these revisions have strengthened the manuscript, and we are grateful for your guidance in improving it. Thank you once again for your time, thoughtful critique, and for helping us refine this work. We look forward to your further comments and hope the revised manuscript meets your expectations.

Response to Reviewer #1’s Comments:

1. Comment: The title does not reflect the entire content of the article; it needs to be reviewed.

Response: Thank you for your valuable feedback regarding the title. We understand the importance of ensuring that it accurately reflects the full scope of the study. In response, we have revised the title to better capture the content and findings of the article: “Prevalence and Determinants of Herbal Medicine Utilization During Childbirth among Women of Reproductive Age in Butiama, Tanzania: A Community-Based Cross-Sectional Study.”

2. Comment: Table 1 should be eliminated because the majority of the information has been repeated in Table 2. The latter is largely sufficient to present the results.

Response: Thank you for your valuable feedback. As per your suggestion, we have eliminated Table 1, as the majority of the information is repeated in Table 2. We agree that Table 2 sufficiently presents the results, and we have made the necessary revisions accordingly.

3. Comment: Please correct in the Page 10, table 1 the value, ‘’never used HM = 165’’ et Utilization of HM during childbirth, No = 164)?

Response: Thank you for pointing this out. We have corrected the values in Table 1 on page 10. The value for "never used HM" has been updated to 165.

4. Comment: Table 1: This question is not correct ‘’ Type of HM used during childbirth’’, because the proposed answers are varied.

Response: Thank you for your feedback. We have addressed the concern regarding the question "Type of HM used during childbirth." In response, we revised the question to specifically inquire about the types of herbal medicine used (as we used during data collection). The revised options now include: Pumpkin roots (Cucurbita pepo), African spider plant (Cleome gynandra), Ginger (Zingiber officinale Roscoe), Concentrated green or black tea (Camellia sinensis), and "Never used herbal medicine." This has been shifted to Table 2.

5. Comment: In Table 2, the sum of the values for all the variables studied does not give 233, which is the total of participants who confirmed the use of plants. All the values in this table must be corrected.

Response: Thank you for your valuable feedback. We have addressed the issue you raised regarding the sum of the values in Table 2 not adding up to 233, the total number of participants who confirmed the use of herbal medicine. The necessary corrections have been made to ensure that all values in the table are accurate and align with the total number of participants.

Response to Reviewer #2’s Comments:

1. Comment: I found that there seems to be several errors in Table 2. For “Age group” the percentages are mixed up (% for “all participants” come under “No”, % for “Yes” belong to “No” etc). This made me do the analysis in SPSS and I get different values for both counts and p-values for most of the factors in the table.

Response: Thank you for your insightful comment. We sincerely appreciate your thorough review of Table 2. Upon reviewing the table, we acknowledge that there were errors in the presentation of the percentages, where the values for "all participants," "No," and "Yes" were incorrectly assigned to the wrong categories. We have corrected these errors and recalculated the relevant counts and p-values to align with the correct data. Thank you again for bringing this to our attention, and we apologize for any confusion caused.

2. Comment: I struggle to see why you have chosen a significance level of 0.25. That is very high. And talking about significance, it is difficult to see what is significantly higher or lower than what.

Response: Thank you for your valuable comment. The significance level of 0.25 was chosen for qualifying variables from the bivariate analysis to be entered into the multivariate analysis, as explained in the methodology section of the paper. This approach follows the guidelines established in reference No. 26 (Martinez et al., 2017), which suggests using a more liberal threshold for inclusion in multivariate models.

3. Comment: For Table 3, many confidence intervals are overlapping – does that mean anything for the results?

Response: Thank you for your comment. We have reviewed the results and adjusted the analysis in the new version of the table to address the issue of overlapping confidence intervals. The updated version ensures a more accurate interpretation of the results.

4. Comment: Table 4 has some numbers in bold and marked with stars – what do those markings refer to? They normally refer to significant values, but if that is the case here, what method is used for the analysis?

Response: Thank you for your valuable feedback. To clarify, we employed descriptive analysis of univariate data to determine the distribution of responses based on the Likert scale. The bolded numbers in Table 4 highlight values of interest, and we have now explained these markings in the sub-text added to the table. This revision ensures that the table is fully self-explanatory.

5. Comment: A definition of “herbal medicine” would be useful. How did you explain to the participants what you were asking about? And is it necessary to make the abbreviation “HM”?

Response: Thank you for your thoughtful comments. We appreciate your suggestion regarding the definition of “herbal medicine.” We have now included a clear definition in the manuscript to ensure that readers understand the term in the context of our study. Additionally, during data collection, we provided participants with a straightforward explanation of herbal medicine, describing it as ‘any plant-based substances, including leaves, roots, flowers, or teas, used to aid in labour and delivery without prescription from formal healthcare providers.’ Regarding the abbreviation “HM,” we recognize that it may not be essential. To enhance clarity and readability, we have decided to remove the abbreviation and instead use the full term “herbal medicine” throughout the manuscript.

6. Comment: Where does reference 1 say that “conventional and alternative therapies contribute to 80% of global healthcare”?

Response: Thank you for your comment. The information regarding the contribution of conventional and alternative therapies to global healthcare is available in the embedded file under the WHO reference. Additionally, we have included two other references that report similar findings. We have also refined the sentence to:

"The World Health Organization (WHO) and other studies report that nearly 50% of people in many industrialized countries use both conventional and alternative therapies, with usage rates reaching up to 80% in numerous developing countries."

7. Comment: You say that “Studies show that herbal medicine during labour induces intense, prolonged contractions…etc.” but which herbal medicines? It cannot be a general statement about herbal medicines.

Response: Thank you for your valuable feedback. We have addressed the concern regarding the specific herbal medicines by clarifying which herbs are associated with the complications mentioned and referencing the relevant studies that detail their effects. The revised text now includes plants used to induce labour, such as oxytocic herbs like the African spider plant (Cleome gynandra), Cissampelos mucronata, Mwanamphepo (Cissus species), Ampelocissus, Ginger and Cyphostemma.

8. Comment: What does the number of modes of administration have to do with the cultural significance of herbal medicine?

Response: Thank you for your comment. We have removed the reference to the number of modes of administration as it was not directly relevant to the cultural significance of herbal medicine.

9. Comment: Your section about training research assistants is rather detailed.

Response: Thank you for your feedback. We have addressed the concern by streamlining the section on training research assistants. The revised version now presents the key points more concisely while retaining all necessary information about the training process.

10. Comment: I guess you mean “Muhimbili University of Health and Allied Sciences.”

Response: Yes, you are correct. We meant "Muhimbili University of Health and Allied Sciences." Thank you for pointing that out.

11. Comment: The response rate is not about how many you aimed to include but about how many you asked to participate versus how many were willing.

Response: Thank you for your comment. We have revised the statement to accurately reflect the response rate. It now reads: "A total of 406 participants were initially approached for the study, with 398 providing informed consent, yielding a response rate of 98.03%." Thank you again for your feedback.

12. Comment: Please reconsider how you write the sentences with numbers and percentages – it does not flow as it is now.

Response: Thank you for your feedback. We have revised the section to ensure the presentation of numbers and percentages flows more smoothly.

13. Comment: I suppose you KNOW how many participants used herbal medicines during childbirth – not “around 233” (which is an approximate number) but “233.” The same goes for “About 272” participants with primary education – it is 272 – not “about.”

Response: Thank you for your observation. You are correct, and we appreciate the clarification. The numbers should indeed be exact rather than approximate. We have revised the text accordingly: "A total of 233 (58.5%) participants used herbal medicine during childbirth."

14. Comment: Why do you mention “roots” and “leaves” but not which roots or leaves? Does this matter? Is it interesting? Why?

Response: Thank you for your comments. We have addressed the concern regarding the question "Type of HM used during childbirth." In response, we revised the question to specifically inquire about the types of herbal medicine used (as we used during data collection). The revised options now include: Pumpkin roots (Cucurbita pepo), African spider plant (Cleome gynandra), Ginger (Zingiber officinale Roscoe), Concentrated green or black tea (Camellia sinensis), and "Never used herbal medicine."

15. Comment: The statement “Women’s desire to maintain marriages,….. Reliance on HM” requires a reference.

Response: Thank you for your comment. We have removed the mentioned statement as it is no longer applicable following the adjustment of our revised analysis. This change ensures that the content aligns with the updated findings and maintains clarity in the discussion.

16. Comment: There is a significant difference between those with no education (reference) and those with primary education, but what can you say about those with primary vs. those with secondary education?

Response: Thank you for your insightful comment. In the refined version of our analysis, the difference between those with primary education and those with secondary education is no longer statistically significant. This change was made after reassessing the data, and we have adjusted the results accordingly. We appreciate your attention to this detail and have made the necessary revisions to reflect the updated findings.

17. Comment: Overlapping confidence intervals – does that matter? Confidence intervals including “1” – does that matter?

Response: Thank you for your question. Overlapping confidence intervals and confidence intervals that include "1" can indeed have implications for the interpretation of the results. We have reviewed and addressed these points in the updated version of the manuscript to ensure the interpretation is clear and accurate.

18. Comment: Please also revise the abstract accordingly.

Response: Thank you for your suggestion. We have revised the abstract accordingly to reflect the changes made in the whole manuscript. Thank you again for your helpful feedback.

---

## [Decision Letter · Decision Letter 1]

16 May 2025

PONE-D-24-53690R1
Prevalence and determinants of herbal medicine utilization during labour among women of reproductive age in Butiama, Tanzania: A community-based cross-sectional study
PLOS ONE

Dear Dr. SICHALWE,

Thank you for submitting your manuscript to PLOS ONE. After careful consideration, we feel that it has merit but does not fully meet PLOS ONE’s publication criteria as it currently stands. Therefore, we invite you to submit a revised version of the manuscript that addresses the points raised during the review process.

We look forward to receiving your revised manuscript.

Kind regards,

Ammal Mokhtar Metwally, Ph.D (MD)

Academic Editor

PLOS ONE

Additional Editor Comments (if provided):

After thorough re-evaluation, we commend the authors for addressing the first round of reviewer comments with care. The study covers an important and under-explored topic in maternal health within Sub-Saharan Africa. However, several methodological, analytical, and presentation-related weaknesses remain, which must be addressed to meet the publication standards of PLOS ONE.

In light of the below concerns, in addition to those of the reviewers. We encourage a major revision that addresses the outlined issues comprehensively.

Key points to be addressed:

1. Statistical Justification and Analysis Approach

The use of a significance threshold of p<0.25 for including variables in multivariate analysis is uncommon and insufficiently justified. While cited literature is acknowledged, this approach may lead to overfitting and misinterpretation.

Table 3 is mislabeled as Poisson regression while results appear to be from logistic regression. This should be corrected, and the statistical method clarified.

Overlapping confidence intervals and the inclusion of 1 in some adjusted odds ratios raise interpretive concerns. A clearer explanation of how these results are interpreted would improve the scientific rigor.

2. Language, Syntax, and Clarity

The manuscript contains awkward phrasing, such as “higher likely,” and inconsistent use of statistical language. For instance, “55.6% higher likely to use herbal medicine” should be rephrased for clarity.

Reporting of percentages should be precise (e.g., avoid “about 233” when the actual number is known).

3. Redundancy and Table Narration

In the results and discussion sections, there is excessive repetition of values already presented in tables. A more interpretive approach that highlights significant findings and their implications is preferred over verbatim restatements.

4. Introduction and Reference Support

Several claims are not adequately referenced, including statistics on global use of traditional medicine. For example, the WHO citation regarding 80% usage needs to be directly traceable and accurately quoted.

The link between cultural beliefs and herbal medicine use is mentioned frequently but would benefit from more empirical support or references.

5. Discussion and Critical Interpretation

The discussion would be strengthened by a more critical appraisal of causal inference limitations, potential confounding, and biases (recall and social desirability).

While cultural influences are acknowledged, the discussion sometimes overgeneralizes and lacks nuance in differentiating between association and causality.

6. Methodological Clarification

Further clarification is required regarding the questionnaire's validation process (e.g., reliability testing metrics such as Cronbach’s alpha were not reported).

The detailed section on training of research assistants, while informative, could be streamlined for conciseness.

7. Terminology and Table Design

Inconsistencies remain in table formatting and terminology, such as mislabeling of columns and unclear indicators of statistical significance (e.g., asterisks not always explained).

The title of Table 3 should accurately reflect the statistical method used.

We appreciate your contribution to this important field of research and look forward to reviewing a significantly improved version of your manuscript,

Reviewers' comments:

Reviewer's Responses to Questions

**Comments to the Author**

1. If the authors have adequately addressed your comments raised in a previous round of review and you feel that this manuscript is now acceptable for publication, you may indicate that here to bypass the “Comments to the Author” section, enter your conflict of interest statement in the “Confidential to Editor” section, and submit your "Accept" recommendation.

Reviewer #2: (No Response)

Reviewer #3: (No Response)

2. Is the manuscript technically sound, and do the data support the conclusions?

Reviewer #2: Yes

Reviewer #3: Partly

3. Has the statistical analysis been performed appropriately and rigorously? 

Reviewer #2: Yes

Reviewer #3: Yes

4. Have the authors made all data underlying the findings in their manuscript fully available?

Reviewer #2: Yes

Reviewer #3: Yes

5. Is the manuscript presented in an intelligible fashion and written in standard English?

Reviewer #2: Yes

Reviewer #3: Yes

6. Review Comments to the Author

Reviewer #2: Thank you for your good replies to all my questions. I only have a few more questions or comments:

Please remember that the number of decimals indicates the accuracy of the number given. You seem to be using one, two or three decimals randomly.

Results:

Lower education p=0,094 is said to be significant – is it?

How do you know that the “adverse effects” mentioned are actually adverse effects of the herbs used? They could be “suspected adverse effects” but do you know for sure that they ARE?

Please consider and discuss confounders: Is “living more than 5 km from nearest clinic” and “attending fewer than 4 ANC visits” the same thing? Or at least interconnected?

Table 3 needs a comment in the subtext explaining the meaning of the * And in the top line you have to put COR [95% CI] and AOR [95% CI].

It is also of interest that Education and Wealth index did not seem to affect the use of herbs.

There are some places in the discussion where I am not sure whether it is your finding/opinion or whether it needs a reference:

- Community education on herbal remedy risks………………..complications.

- Additionally, involving community leaders, ……………..maternal care.

- Enhancing community education on the safe use…….informed choices.

I agree that the “community based” approach is an advantage. It is not very well known in Europe so a comment on WHY it is a strength would be good.

Reviewer #3: Dear corresponding author

The topic is relevant and interesting; however, the authors should address the following points to improve the manuscript:

1. The statement of the study’s purpose at the end of the introduction (last three lines, reference 23) is confusing and appears redundant. This phrasing is more appropriate for a research proposal rather than a journal article introduction. I recommend deleting this sentence and instead refining the purpose as stated earlier in the introduction.

2. In the Materials and Methods section, given that this is a cross-sectional (descriptive) study, the use of terms such as "independent variable" and "dependent variable" is inappropriate. Cause-and-effect relationships are not applicable in descriptive studies but rather in cohort, case-control, or intervention designs. It would be better to refer to all variables as contextual or descriptive.

3. The description of obtaining written informed consent from unmarried participants under 18 years old is unclear. Since the inclusion criterion requires prior childbirth experience, it is questionable whether unmarried individuals under 18 would meet this criterion, especially considering cultural contexts such as those in many Middle Eastern countries. The authors should clarify this point in the manuscript.

4. The study population consists of women who gave birth between 2021 and 2024, with data collected from June to July 2024. Given this timeline, recall bias is a potential concern, particularly regarding herbal medicine use during childbirth. The authors should discuss how recall bias was addressed or minimized and provide more information on the validity of the questions related to herbal medicine use.

5. The distinction between herbal medicines and medicinal plants needs clarification. Herbal medicines typically refer to products that have undergone formulation processes (e.g., medicines, syrups, creams), whereas medicinal plants are raw or minimally processed. The authors should specify their definition and approach regarding this distinction.

6. The limitations of the study should be presented before the conclusions. Furthermore, the conclusions and recommendations should be aligned strictly with the study’s actual findings.

Overall, the article is suitable for publication after major revisions addressing the above points.

Thank you for considering these comments.

Sincerely,

7. PLOS authors have the option to publish the peer review history of their article (what does this mean?). If published, this will include your full peer review and any attached files.

Reviewer #2: No

Reviewer #3: No

---

## [Author Response · Author response to Decision Letter 2]

24 Jun 2025

We sincerely thank the reviewers for their thoughtful and constructive feedback. We have carefully addressed each of the comments and made the necessary revisions to the manuscript accordingly. A detailed point-by-point response to all reviewer comments has been thoroughly prepared and uploaded for your consideration (uploaded under Responses to Reviewers’ Comments).

We greatly appreciate your time and valuable insights, which have significantly contributed to improving the quality of our manuscript.

Responses to Reviewer #2:

Decimal Consistency and P-value Accuracy – We standardized all statistical values to three decimal places. Education was not statistically significant in the revised version.

Adverse Effects Clarification – We now refer to them as self-reported suspected adverse effects and acknowledge the limitations of attributing causality.

Confounding Factors – We acknowledged and discussed the potential confounding between distance to health facilities and ANC visits in the discussion section.

Table 3 Revision – The table has been updated with proper headers (COR [95% CI], AOR [95% CI]) and explanatory notes.

Strength of Community-Based Approach – We expanded the discussion to explain the unique strengths of using a community-based design in the Tanzanian context.

Responses to Reviewer #3:

Study Purpose – The final paragraph of the introduction was rephrased and simplified to better reflect a journal-style aim.

Terminology in Methods – We replaced "independent" and "dependent" variables with "contextual" or "descriptive" variables, as appropriate for a cross-sectional study.

Consent Clarification – We clarified that participants under 18 were either married or in unions, in line with cultural norms, and that consent procedures followed ethical guidelines.

Recall Bias – The data collection timeline (2023–2024) was clarified to minimize concerns about recall bias.

Definition of Herbal Medicines – We clarified our focus on raw or minimally processed medicinal plants in the operational definitions and across the manuscript.

Structure and Recommendations – We repositioned the limitations section before the conclusions and revised the recommendations to align strictly with the study findings.

We hope the revised manuscript and responses meet your expectations and are now suitable for publication. We remain grateful for the opportunity to revise and resubmit our work.

---

## [Decision Letter · Decision Letter 2]

23 Jul 2025

PONE-D-24-53690R2
Prevalence and determinants of medicinal plants utilization during labour among women of reproductive age in Butiama, Tanzania: A community-based cross-sectional study
PLOS ONE

Dear Dr. Sichalwe,

Thank you for submitting your manuscript to PLOS ONE. After careful consideration, we feel that it has merit but does not fully meet PLOS ONE’s publication criteria as it currently stands. Therefore, we invite you to submit a revised version of the manuscript that addresses the points raised during the review process.

We look forward to receiving your revised manuscript.

Kind regards,

Ammal Mokhtar Metwally, Ph.D (MD)

Academic Editor

PLOS ONE

Journal Requirements:

Reviewers' comments:

Reviewer's Responses to Questions

**Comments to the Author**

1. If the authors have adequately addressed your comments raised in a previous round of review and you feel that this manuscript is now acceptable for publication, you may indicate that here to bypass the “Comments to the Author” section, enter your conflict of interest statement in the “Confidential to Editor” section, and submit your "Accept" recommendation.

Reviewer #2: (No Response)

Reviewer #3: All comments have been addressed

2. Is the manuscript technically sound, and do the data support the conclusions?

Reviewer #2: Yes

Reviewer #3: Partly

3. Has the statistical analysis been performed appropriately and rigorously? 

Reviewer #2: I Don't Know

Reviewer #3: Yes

4. Have the authors made all data underlying the findings in their manuscript fully available?

Reviewer #2: Yes

Reviewer #3: Yes

5. Is the manuscript presented in an intelligible fashion and written in standard English?

Reviewer #2: Yes

Reviewer #3: Yes

6. Review Comments to the Author

Reviewer #2: Dear Author,

Thank you for the replies to my questions.

Please check once again for spelling mistakes.

Please be consistent: write all Latin names in italics (f.x. page 3 and Table 2, but could be anywhere), English/African names should NOT be in italics (f.x. page / "Main outcome variables", but could be other places as well).

"...pregnancy and or labour" should be "pregnancy and/or labour".

All tables and figures should be understandable "in themselves".

Table 2: "suspected adverse effects" - not only in text but in the table too.

Page 12 "Predictors of ....": numbers of decimals says something about how exact a number is - please be consistent and perferably use no more than one decimal when you say that "X had x.y times higher odds than Y" (to be "1.823 times more likely to use medicinal plants" is overdoing it - 1.8 would be fine). And please change in the abstract too.

Table 3: what does the * refer to? Needs to be explained below the table. And I still think it is incorrect to say that "distance" and "number of visits" are both significant as they are most probably interrelated. A comment would be good.

Page 14: "Perceptions and beliefs...": I don't see 48.7 in the table? Maybe say that"... 48.7% strongly agreed or agreed...". The same goes for the other numbers.

Table 4: the sub-text about "bolded values" is unclear - please rephrase. Item 2 and 3:" ... medicinal plants are ...."

Reviewer #3: Dear author

Thank you for giving me another opportunity to review the article titled " Prevalence and determinants of medicinal plants utilization during labour amongwomen of reproductive age in Butiama, Tanzania: A community-based cross-sectional study". “PONE-D-24-53690R2” I have carefully read the previous reviewers' comments, examined the authors' responses, and reviewed the revised manuscript. While the authors have made significant improvements, some ambiguities still require further clarification and revision, as outlined below.

1. Given that participants were asked about their use of medicinal plants "during labour and delivery," I would like to clarify how this period was defined for the participants to avoid confusion with the pregnancy period. Could you please provide an operational definition of "labour and delivery" as used in your study? Specifically, from what point in time was this period considered to begin and end. The use of medicinal plants for labor induction, labor acceleration, or other purposes during pregnancy among women with previous pregnancy and childbirth experience is still unclear. Given that participants were asked about the purpose of use, we kindly request that you define and clarify the operational meaning of "labor induction and acceleration" in the Materials and Methods section of the manuscript.

2. The sample size should be clearly specified in Table 3.

3. What are the study limitations? Please address potential issues such as participants possibly misunderstanding the questions, external factors like time constraints, resource availability, or local conditions, variations in sampling or data collection methods, and even the sample size itself.

4. The data collection method needs to be described more clearly and with greater detail. Considering the participants’ educational levels and literacy, was a uniform data collection method applied to all? Please provide further explanation

5. Regarding the reviewer’s previous comment: "A definition of 'herbal medicine' would be useful. How did you explain what you were asking about to the participants? And is it necessary to abbreviate it as 'HM'?" — and the authors’ response, the explanation provided seems more useful for the readers of the article rather than addressing how the concept was explained to the participants. Although the authors gave good explanations to help participants understand, these may still be insufficient, as participants might have varied interpretations. If this is the case, it should be acknowledged as a limitation of the study.

6. Regarding the validity of the questionnaire on participants’ perceptions and beliefs about the use of medicinal plants during childbirth, which was evaluated by five experts in public health, maternal and child health, and linguistics, the reported content validity index (CVI) of 0.90 for each item and over 0.90 overall (0.93) requires further clarification. Specifically, how was the 3% increase in the overall CVI achieved?

Additionally, considering the internal consistency of the questions, with a Cronbach’s alpha coefficient of 0.84, it is expected that the results section—especially the part related to Table 4—will include a classification of participants’ beliefs into levels such as weak, average, and good, or any other categorization the authors find appropriate. This classification would enhance the discussion and interpretation of the findings.

Best regards

7. PLOS authors have the option to publish the peer review history of their article (what does this mean?). If published, this will include your full peer review and any attached files.

Reviewer #2: No

Reviewer #3: No

---

## [Author Response · Author response to Decision Letter 3]

25 Jul 2025

Dear Reviewers,

We sincerely thank you for your thoughtful and constructive comments on our manuscript. Your detailed feedback has significantly improved the clarity, accuracy, and quality of our work. We have carefully addressed all the comments raised by each reviewer, and all corresponding changes have been made in the revised manuscript. (complete responses uploaded as S1 Rebuttal letter to Reviewers)

Below is a brief summary of the major revisions:

We thoroughly reviewed the manuscript for spelling errors and ensured consistency in italicization, using italics only for Latin (scientific) names.

We corrected terminology and expressions for clarity (e.g., changed "pregnancy and or labour" to "pregnancy and/or labour" or Labour and/or delivery).

Tables have been revised to be self-explanatory, with appropriate footnotes added (e.g., Table 3 asterisk explanation).

Decimal precision has been standardized (limited to one decimal place) across the manuscript, including in the abstract.

 We provided clear operational definitions for key concepts, such as “labour and delivery,” “labour induction,” and “labour acceleration,” as used during data collection.

 The Limitations section has been expanded to acknowledge possible misinterpretations by participants and external challenges affecting data quality.

 We clarified the data collection process, emphasizing the use of trained interviewers and a uniform Kiswahili questionnaire to accommodate all literacy levels.

 We removed the abbreviation “HM” and instead consistently used the full term “medicinal plants.”

 The content validity index (CVI) and Cronbach’s alpha were explained, and participant beliefs were further categorized to enhance the discussion of Table 4.

We are grateful for your feedback, which has helped us refine the manuscript for greater scientific rigor and readability. Please do not hesitate to let us know if any further clarification is needed.

Best regards,

Corresponding Author on behalf of all co-authors

---

## [Decision Letter · Decision Letter 3]

3 Sep 2025

PONE-D-24-53690R3
Prevalence and determinants of medicinal plants utilization during labour among women of reproductive age in Butiama, Tanzania: A community-based cross-sectional study

PLOS ONE

Dear Dr. Sichalwe,

Thank you for submitting your manuscript to PLOS ONE. After careful consideration, we feel that it has merit but does not fully meet PLOS ONE’s publication criteria as it currently stands. Therefore, we invite you to submit a revised version of the manuscript that addresses the points raised during the review process.

Form letter of Editorial (SEE Framework)

Dear Authors,

Thank you for the opportunity to review your revised manuscript - "Prevalence and determinants of medicinal plants utilization during labour among women of reproductive age in Butiama Tanzania: A community based, cross-sectional study" (PONE-D-24-53690R3). I read your manuscript carefully to provide you with feedback, and I have organized the comments to you based on the SEE (Statement-Explanation-Example) framework.

Title and Abstract

Statement: In general, the title and abstract were sufficiently informative, with the caveat that the abstract could be tightened up.

Explanation: The authors repeated "labour and/or delivery" to a fault, which overall resulted in a wordy paper, in part because the ambiguous end to their text.

Example: The summary sentence for this section could have simply been: "Over fifty percent of participants reported utilizing medicinal plants during childbirth, with various socio-economic status and health care access factors indicating that education and interventions targeted around medicinal plants would be advantageous.

Introduction

Statement: Overall, I thought the introduction was decent, but a bit bogged-down with bloated content.

Explanation: The sheer number of plants (some with awkward language (helpful info overall), broke my flow and allowed me to disengage with it.

Example: I think you can talk about prevalence and risk factors more than you can plant descriptions. You could have situated plant examples in the Discussion.

Methods

Statement: Overall, I thought the reporting approach was strong, and my understanding of the reporting was clear, but there were areas that were unnecessary details that went above and beyond relevance to the study.

Explanation: The ethical considerations and validation testing were aptly described, but it seems the authors devoted a substantial amount of detail to more lengthy operational definitions of "labour induction/acceleration".

Example: The broadness of "labour induction/acceleration", or detailed specificities could have been included in the results section with the other operational definitions in a paragraph of precise text.

Results

Statement: Overall the results were clear, with identified predictors and estimates of prevalence, but could have been expressed more efficiently.

Explanation: Mostly the results section was legibility by primarily expressing some details in both the tables as well as in the text.

Example: Highlight the key predictors and odds ratios as indicated for example "Women who used less than 4 ANC visits (AOR=0.556, p=0.006)" were more likely to use medicinal plants. The participants were asked every percentage?

Discussion

Statement: The discussion is thorough and fairly contextualized, perhaps too extensive.

Explanation: Comparisons from region to region were useful for comparisons, however sometimes they pulled away the uniqueness of this study.

Example: It would help to remain focused on the three themes, explain socio-economic / cultural factors, and access to health care and ANC utilizations, and be explicit regarding the unique findings for Butiama district.

Limitations

Statement: The authors are honest limitations, although overall fairly lengthy.

Explanation: Some details (environmental conditions) fails to be relevant overall your manuscript.

Example: Can summarize: The main limitations were cross-sectional and recall bias, and the potential misinterpretation of 'medicinal plants'even with clear explanation from interviewer.

Conclusions and Recommendations

Statement: The conclusion was solid, with some degree of verbal blurring with the recommendations.

Explanation: It would be useful to gain a little more distance from our findings to make suggestions, but the recommendations would still be sectioned distinctly from the conclusions.

Example: Conclude the statements summarized in the conclusion with, "This study brought timely attention to the importance of speaking about the use of medicinal plants as we provide maternal health services." and just keep the recommendations as a section.

References

Statement: Overall the references are appropriate to the context of the research and largely current, however some were lesser sources.

Explanation: In general, less of the websites or unlabelled attribution would be helpful in not pulling from potential sources.

Example: Source potential from other peer-reviewed literature (i.e., the systematic reviews that review herbal medicine use during childbirth) or methodologies.

Overall, the manuscript is methodologically rigorous, ethically sound and a contribution to the knowledge in maternal health (and maternal health research) literature base in sub-Saharan Africa.

We look forward to receiving your revised manuscript.

Kind regards,

Ammal Mokhtar Metwally, Ph.D (MD)

Academic Editor

PLOS ONE

Journal Requirements:

Reviewers' comments:

Reviewer's Responses to Questions

**Comments to the Author**

1. If the authors have adequately addressed your comments raised in a previous round of review and you feel that this manuscript is now acceptable for publication, you may indicate that here to bypass the “Comments to the Author” section, enter your conflict of interest statement in the “Confidential to Editor” section, and submit your "Accept" recommendation.

Reviewer #2: (No Response)

Reviewer #3: All comments have been addressed

2. Is the manuscript technically sound, and do the data support the conclusions?

Reviewer #2: Yes

Reviewer #3: Yes

3. Has the statistical analysis been performed appropriately and rigorously? 

Reviewer #2: I Don't Know

Reviewer #3: Yes

4. Have the authors made all data underlying the findings in their manuscript fully available?

Reviewer #2: Yes

Reviewer #3: Yes

5. Is the manuscript presented in an intelligible fashion and written in standard English?

Reviewer #2: Yes

Reviewer #3: Yes

6. Review Comments to the Author

Reviewer #2: Dear Authors,

Now I only have a very few comments tot he manuscript, as you have done a good job revising it.

Please check for a few more "and or" which should be "an/or".

First line in Introduction: The World health organisation (WHO) and various (not "other" as WHO is not a study).

Same page, line 11: ".... and increased risk of cesarean section" (or increased number of cesarean...)

In Materials and methods, Sampling technique, First stage you talk about "wards" - is that the right word? Or is it "clinics"?

Main outcome variables, you mention "pumpkin roots" - are the roots actually used? Or the pumpkin it self or the seeds?

Pre-testing of data collection instruments: the questionnaire was administered "orally" - not "verbally".

Table 2: Here it still says "to enhance labour" (should be "hasten" as in the text above the table).

With those minor changes, I think your paper is ready for publishing.

Reviewer #3: Dear Author

Please ensure all abbreviations are spelled out in full when first used in the abstract, main text, or footnotes for clarity.

Best regard

7. PLOS authors have the option to publish the peer review history of their article (what does this mean?). If published, this will include your full peer review and any attached files.

Reviewer #2: No

Reviewer #3: No

---

## [Author Response · Author response to Decision Letter 4]

5 Sep 2025

We sincerely thank you and the reviewers for the constructive and insightful feedback provided. We have carefully considered all comments and revised the manuscript accordingly. In the attached document, we provide a detailed, point-by-point response to each comment raised by the reviewers, outlining the changes made and justifications where applicable.

We believe these revisions have significantly improved the clarity, rigor, and overall quality of the manuscript.

We appreciate the time and effort invested by the reviewers and the editorial team in evaluating our work and look forward to your favorable consideration for publication. (submitted as a Response to Reviewers)

---

## [Decision Letter · Decision Letter 4]

28 Sep 2025

Prevalence and determinants of medicinal plants utilization during labour among women of reproductive age in Butiama, Tanzania: A community-based cross-sectional study

PONE-D-24-53690R4

Dear Dr. Sichalwe,

We’re pleased to inform you that your manuscript has been judged scientifically suitable for publication and will be formally accepted for publication once it meets all outstanding technical requirements.

Kind regards,

Ammal Mokhtar Metwally, Ph.D (MD)

Academic Editor

PLOS ONE

Additional Editor Comments (optional):

Reviewers' comments:

Reviewer's Responses to Questions

**Comments to the Author**

1. If the authors have adequately addressed your comments raised in a previous round of review and you feel that this manuscript is now acceptable for publication, you may indicate that here to bypass the “Comments to the Author” section, enter your conflict of interest statement in the “Confidential to Editor” section, and submit your "Accept" recommendation.

Reviewer #2: All comments have been addressed

2. Is the manuscript technically sound, and do the data support the conclusions?

Reviewer #2: (No Response)

3. Has the statistical analysis been performed appropriately and rigorously? 

Reviewer #2: (No Response)

4. Have the authors made all data underlying the findings in their manuscript fully available?

Reviewer #2: (No Response)

5. Is the manuscript presented in an intelligible fashion and written in standard English?

Reviewer #2: (No Response)

6. Review Comments to the Author

Reviewer #2: (No Response)

7. PLOS authors have the option to publish the peer review history of their article (what does this mean?). If published, this will include your full peer review and any attached files.

Reviewer #2: No

---

## [Editor Report · Acceptance letter]

PONE-D-24-53690R4

PLOS ONE

Dear Dr. Sichalwe,

I'm pleased to inform you that your manuscript has been deemed suitable for publication in PLOS ONE. Congratulations! Your manuscript is now being handed over to our production team.

Kind regards,

on behalf of

Professor Ammal Mokhtar Metwally

Academic Editor

PLOS ONE